# Placental transfer of Letermovir & Maribavir in the *ex vivo* human cotyledon perfusion model. New perspectives for *in utero* treatment of congenital cytomegalovirus infection

Valentine Faure Bardon[1,2], Gilles Peytavin[3], Minh Patrick Lê[3], Tiffany Guilleminot[2,4], Elisabeth Elefant[5], Julien Stirnemann[1,2], Marianne Leruez-Ville[2,4], Yves Ville[1,2]*

1 APHP, Fetal Medicine and Obstetric Department, Necker-Enfants Malades Hospital, Paris, France,
2 EHU7328, IMAGINE Institute, University Paris Descartes, Paris, France, 3 APHP, Pharmacology & Toxicology Laboratory and IAME, Inserm UMR 1137, Université Paris, Paris, France, 4 APHP, Virology Laboratory, National Reference Laboratory for congenital CMV, Necker-Enfants Malades Hospital, Paris, France, 5 CRAT, Reference Centre on Teratogenic Agents, APHP, Trousseau Hospital, Paris, France

* yves.ville@aphp.fr

## Abstract

### Background

Congenital cytomegalovirus infection can lead to severe sequelae. When fetal infection is confirmed, we hypothesize that fetal treatment could improve the outcome. Maternal oral administration of an effective drug crossing the placenta could allow fetal treatment. Letermovir (LMV) and Maribavir (MBV) are new CMV antivirals, and potential candidates for fetal treatment.

### Methods

The objective was to investigate the placental transfer of LMV and MBV in the *ex vivo* method of the human perfused cotyledon. Term placentas were perfused, in an open-circuit model, with LMV or MBV at concentrations in the range of clinical peak plasma concentrations. Concentrations were measured using ultraperformance liquid chromatography coupled with tandem mass spectrometry. Mean fetal transfer rate (FTR) (fetal (FC) /maternal concentration), clearance index (CLI), accumulation index (AI) (retention of each drug in the cotyledon tissue) were measured. Mean FC were compared with half maximal effective concentrations of the drugs ($EC_{50(LMV)}$ and $EC_{50(MBV)}$).

### Results

For LMV, the mean FC was (± standard deviation) $1.1 \pm 0.2$ mg/L, 1,000-fold above the $EC_{50(LMV)}$. Mean FTR, CLI and AI were $9 \pm 1\%$, $35 \pm 6\%$ and $4 \pm 2\%$ respectively. For MBV, the mean FC was $1.4 \pm 0.2$ mg/L, 28-fold above the $EC_{50(MBV)}$. Mean FTR, CLI and AI were $10 \pm 1\%$, $50 \pm 7\%$ and $2 \pm 1\%$ respectively.

**Data Availability Statement:** All relevant data are within the manuscript and its Supporting Information files.

**Funding:** This work was supported by the University Paris Descartes.

**Competing interests:** - Valentine Faure Bardon: No conflict - Gilles Peytavin has received travel grants, consultancy fees, honoraria or study grants from various pharmaceutical companies, including Bristol-Myers Squibb, Gilead Sciences, Janssen, Merck and ViiV Healthcare. - Minh Patrick Lê has received travel grants from Bristol-Myers Squibb and Janssen. - Tiffany Guilleminot: No conflict - Elisabeth Elefant: No conflict - Julien Stirnemann: No conflict - Marianne Leruez-Ville declares receiving financial support for meeting expenses from BioMerieux outside the submitted work. - Yves Ville: reports non-financial support from Ferring SAS, non-financial support from Siemens Health Care, non-financial support from GE medical, outside the submitted work. This does not alter our adherence to PLOS ONE policies on sharing data and materials.

## Conclusions

Drugs' concentrations in the fetal side should be in the range for in utero treatment of fetuses infected with CMV as the mean FC was superior to the $EC_{50}$ for both molecules.

## Introduction

Congenital cytomegalovirus (cCMV) infection is a major cause of neurological and sensory impairment in children. The overall CMV birth prevalence estimate was 0.7%, based on 15 studies with a total of 117,986 infants screened [1]. Vertical transmission occurs either after maternal primary infection or after a secondary infection (reactivation of the endogen CMV strain or reinfection with a new CMV strain). CMV is a viraemic herpes virus reaching to the fetus through the placenta, which acts as a barrier against CMV but also as a reservoir where the virus replicates. All organs of an infected fetus could be affected including the developing brain in the worst cases, leading to a poor prognosis [2].

Currently, cCMV infection is not officially screened for in any country although serological screening is routinely offered to pregnant women in many developed countries. The diagnosis, during pregnancy, can be suspected in women showing seroconversion or in case of compatible features on fetal ultrasound. The diagnosis of fetal infection is made by the detection of viral DNA by polymerase chain reaction (PCR) in the amniotic fluid following amniocentesis. One can hypothesize that prompt initiation of an effective treatment in cases of proven fetal infection could reduce both placental and fetal damages.

To date, antiviral fetal treatment has shown a plausible benefit using valaciclovir, but the use of potent anti-CMV drugs in pregnancy is a debated issue [2,3]. (Val)ganciclovir, cidofovir, and foscarnet are highly genotoxic in vitro and are not approved for use during pregnancy. In addition, maternal toxicities could be significant. Valacyclovir is less effective against CMV, in vitro, but has the best safety profile of the anti-CMV, with no increased risk of birth defects in the offspring of thousands of women exposed during pregnancy [4]. All these drugs inhibit the viral replication by interacting with virally encoded DNA polymerase. Letermovir (LMV) and Maribavir (MBV) are newly announced specific and effective anti-CMV drugs acting through new antiviral mechanisms likely to alleviate the side effects and toxicity described above. Because clinical trials are challenging in pregnant women, preclinical approaches are warranted. Our objective was to investigate the placental transfer of MBV and LMV in the *ex vivo* human cotyledon perfusion model which is a well validated model to study placental transfer of drugs [5,6].

## Materials and methods

### Placentas

Placentas were collected in the maternity ward of Necker Hospital (Paris, France) after uncomplicated pregnancies and term deliveries. Women were informed of the use of their placentas and gave written consents. Procedures were approved by the ethics committee (Agence de Biomédecine, approval: PFS 15–009). General clinical characteristics are presented in Table 1. The perfusion experiment was initiated immediately on site and completed within two hours after delivery. Time from delivery to processing was summarized in Table 2.

First the placenta was examined macroscopically, and an isolated cotyledon was chosen for its integrity. In the fetal side of this cotyledon, the chorionic plate artery and its associated vein

**Table 1. Clinical characteristics.**

| | Letermovir | | | | Maribavir | | | |
|---|---|---|---|---|---|---|---|---|
| | **Mean** | **SD** | **Range** | | **Mean** | **SD** | **Range** | |
| | | | **min** | **max** | | | **min** | **max** |
| **Gravidity** | 2,4 | 0,9 | 1 | 3 | 1,7 | 0,6 | 1 | 2 |
| **Parity** | 2,2 | 0,8 | 1 | 3 | 1,3 | 0,6 | 1 | 2 |
| **Gestational age (weeks)** | 38,4 | 1,1 | 37 | 40 | 39,3 | 1,5 | 38 | 41 |
| **Maternal age** | 32 | 3 | 30 | 37 | 32,0 | 1,7 | 31 | 34 |
| **Birth weight (gram)** | 3292 | 207 | 3085 | 3605 | 3205 | 213 | 3000 | 3425 |
| **Prenatal medications** | None N = 5 | | | | None N = 3 | | | |
| **Drugs (including tobacco)** | None N = 5 | | | | None N = 3 | | | |
| **Previous prenatal admission** | None N = 5 | | | | None N = 3 | | | |
| **Blood pressures <140/90 mmHg** | All N = 5 | | | | All N = 3 | | | |
| **Gestational diabetes** | None N = 5 | | | | None N = 3 | | | |
| **Antibiotics in labor** | None N = 5 | | | | None N = 3 | | | |
| **Beta strep status** | None N = 5 | | | | None N = 3 | | | |
| **Antenatal steroids** | None N = 5 | | | | None N = 3 | | | |
| **Magnesium sulfate** | None N = 5 | | | | None N = 3 | | | |
| **Anesthesia** | Epidural N = 5 | | | | Epidural N = 3 | | | |
| **Cervical ripening agent** | None N = 5 | | | | None N = 3 | | | |
| **Delivery mode** | Vaginal N = 3 hours of labor = 5h +/- 6.1h | | | | Vaginal N = 1 hours of labor = 5h | | | |
| | C-section, Repeat, no labor: n = 2 | | | | C-section, Repeat, no labor: n = 2 | | | |
| **Maternal Oxygen at delivery** | None N = 5 | | | | None N = 3 | | | |
| **Cotyledon weight (for accumulation index) mean +/SD** | 258 g +/- 76 g | | | | 250 g +/- 53 g | | | |
| **Baby's sex** | Male N = 1 | | | | Male N = 2 | | | |
| | Female N = 4 | | | | Female N = 1 | | | |

were identified and cannulated with two needles. Then, the placenta was placed in the perfusion chamber with the maternal side up. The perfused cotyledon was easily distinguishable while starting to be white due to the fetal perfusion. Finally, we perfused the intervillous space on the maternal side, with two needles in the middle of the whitened cotyledon. Maternofetal circulation was then established.

## Drugs

Powder form of the drugs (for research use only) were bought from MedChem Express (NJ 08852, USA): LMV (CAS N°917389-32-3, purity 98.87%) and MBV (CAS N°176161-24-3, purity 98.69%).

**Table 2. Mean maternal and fetal parameters during the validated procedures.**

| | Letermovir | Maribavir |
|---|---|---|
| Fetal pressure (mmHg) | 52 +/- 11 | 54 +/- 8 |
| Maternal pressure (mmHg) | 14 +/- 6 | 17 +/- 3 |
| Fetal pH | 7.22 +/- 0.02 | 7.23 +/- 0.02 |
| Maternal pH | 7.42 +/- 0.02 | 7.41 +/- 0.03 |
| Fetal temperature (Celsius) | 37.3 +/- 0.2 | 37.1 +/0.3 |
| Maternal temperature (Celsius) | 37.2 +/- 0.4 | 37.2 +/- 0.2 |
| Delivery to processing (mins) | 20 +/- 5 | 15 +/- 5 |

## Cotyledon perfusion

Placentas were perfused in an open circuit model, according to a method described by Schneider et al [7]. Maternal and fetal solutions were prepared with Earle medium (Earle's Balanced Salt 10 X, US Biological, Salem, MA, USA) plus 2 g/Liter of human serum albumin (Ydralbumin 20%, LFB-Biomedicament, Courtaboeuf, France). In the maternal solution, we added the study drug (LMV or MBV) prepared with 2 mL of DMSO (Merck, Germany) as a solvent. The dilution was calculated with target concentrations for the active drug within the range of clinical peak concentrations in plasma ($C_{max}$). For a 480 mg dose of LMV, $C_{max(LMV)}$ = 13 mg/L [8], for a 400 mg dose of MBV, $C_{max(MBV)}$ = 16.1 mg/L [9]. We also added 20 mg/L of antipyrine (Merck, Germany) in the maternal compartment. This is an inert molecule that diffuses freely through the placenta, and acts a marker to validate the cotyledon's viability throughout the experiment [10]. Sodium hydroxide was used to adjust the pH in both compartments to reach a fetal pH of 7.20 and a maternal pH of 7.40. Each circuit, maternal and fetal, included: a warming bath where the solution was maintained at 37°C, a peristaltic pump (running at 6 and 12 mL/min in the fetal and maternal sides respectively), a flowmeter. Two manometers were connected, one in the maternal side and one in the fetal artery side. Perfusion pressure and its stability were then measured during controlled experiment. The mean perfusion pressures values in the fetal and maternal side were, under 65 mmHg and under 20 mmHg respectively during all procedures, (Table 2).

Parameters (temperatures, pressures, pH) were recorded and data were summarized in Table 2. Samples were collected every 5 minutes to determine the antiviral drug and antipyrine concentrations in the fetal vein side and in the maternal compartment. The total perfusion duration was 90 minutes. Samples were then stored at -80°C until analysis. The concentrations were measured using ultraperformance liquid chromatography coupled with tandem mass spectrometry (Waters Xevo UPLC-TQD, Milford, MA, USA).

Four different parameters were looked for, all calculated at steady state: mean fetal concentration (FC), mean maternal concentration (MC), fetal transfer rate (FTR) calculated as the ratio of FC to MC, clearance index (CLI ) which calculated as the ratio of the FTR of the study drug to the FTR of the control (= antipyrine). An antipyrine FTR over 20% was required to validate each experiment [6]. In addition, for LMV and MBV, the results of the mean FC were analyzed in conjunction with their half maximal effective concentrations ($EC_{50}$), as the $EC_{50}$ is a marker of the drug's potency. According to the literature, $EC_{50}$ of LMV = 2.1 nM = 1.2 x$10^{-3}$ mg/L [8] and $EC_{50}$ of MBV = 0.12 µM = 0.05 mg/L [11].

Furthermore, at the end of each perfusion, a piece of tissue was collected, weighted (approximately 250 mg by sample, mean weight is in Table 1) and crushed. Two different samples from the same cotyledon were collected by placenta. Then, the drug was extracted from the cotyledon and its concentration was determined using ultra-performance liquid chromatography coupled with tandem mass spectrometry. Retention of each drug in the placental cotyledon tissue was determined, and expressed as an accumulation index (AI), defined by the ratio of the concentration of the study drug in the cotyledon (ng/mg) multiplied by the tissue density (1.04 g/mL) divided by the mean maternal drug concentration of the relevant experiment.

## Results

### Letermovir (Table 3, Fig 1A)

Among the 10 experiments of placental perfusion, 5 were validated with a mean ± standard deviation (SD) antipyrine FTR of 31 ± 2%.

**Table 3. Mean maternal and fetal parameters for Letermovir and Maribavir at steady state.**

|  | Mean FC ± SD (mg/L) | Mean MC ± SD (mg/L) | Mean FTR ± SD | Mean CLI ± SD | Mean concentration of the study drug in the cotyledon (ng/mg) | Mean AI ± SD |
|---|---|---|---|---|---|---|
| **LMV** | 1.1 ± 0.2 | 12.5 ± 0.7 | 0.09 ± 0.01 | 0.35 ± 0.06 | 0.62 ± 0.32 | 0.04 ± 0.02 |
| **MBV** | 1.4 ± 0.2 | 14.3 ± 0.5 | 0.1 ± 0.01 | 0.5 ± 0.07 | 0.25 ± 0.10 | 0.04 ± 0.01 |

LMV: Letermovir, MBV: Maribavir, FC: fetal concentration, MC: maternal concentration, FTR: fetal transfer rate, CLI: clearance index, AI: accumulation index, SD: standard deviation

In these validated procedures, the system reached steady state at $T_{30\ min}$. The concentration in the maternal compartment was (mean ± SD) $MC_{LMV} = 12.5 \pm 0.7$ mg/L. The concentration in the fetal vein was: $FC_{LMV} = 1.1 \pm 0.2$ mg/L [8]. As the $EC_{50}$ of LMV = 1.2 x$10^{-3}$ mg/L [8], $FC_{LMV}$ was approximately 1,000 fold above the $EC_{50}$. The mean FTR was $FTR_{LMV} = 9 \pm 1\%$ and the mean CLI was $CLI_{LMV} = 35 \pm 6\%$. The mean AI was $AI_{LMV} = 4 \pm 2\%$.

### Maribavir (Table 3, Fig 1B)

Among the 8 experiments of placental perfusion, 3 were validated with a mean ± standard deviation (SD) antipyrine FTR of 21 ± 1%. The system reached steady state at $T_{20min}$.

In these validated procedures, the concentration in the maternal compartment was (median ± SD) = 14.3 ± 0.5 mg/L. The concentration in the fetal vein was $FC_{MBV} = 1.4 \pm 0.2$ mg/L. As the $EC_{50}$ of MBV = 0.05 mg/L [11], $FC_{MBV}$ was approximately 28 fold above the $EC_{50}$. The mean FTR was $FTR_{MBV} = 10 \pm 1\%$ and the mean CLI was $CLI_{MBV} = 50 \pm 7\%$. The mean AI was $AI_{MBV} = 2 \pm 1\%$.

## Discussion

In this *ex vivo* human cotyledon perfusion model, LMV and MBV crossed the placenta at a low to moderate rate ($FTR_{LMV} = 9 \pm 1\%$ and $FTR_{MBV} = 10 \pm 1\%$) and the mean concentration in the fetal compartment was superior to the $EC_{50}$ for both molecules. Small parts of the perfused concentrations accumulated in the placental tissue. As the CMV replicates in the placenta [2], the accumulation of the drugs in this reservoir might contribute to the antiviral efficacy. This opens new perspectives for in utero treatment of infected fetuses using either of these 2 candidate-drugs.

*In vivo*, over 24 hours, the concentration of the antiviral will vary in plasma between the clinical peak concentration ($C_{max}$) and the minimal concentration ($C_{min}$). To treat the fetus efficiently, fetal concentrations of the antiviral should be superior to the $EC_{50}$ of the drug over the whole day. Our results suggest that this would be the case since fetal $C_{max}$ and $C_{min}$ would be above the $EC_{50}$ of both drugs. In this study, the drugs were used within the range of the $C_{max}$ and we showed that the mean concentration in the fetal compartment was superior to the $EC_{50}$ for both anti-CMV drugs ($FC_{LMV} \approx 1,000\ EC_{50}$ and $FC_{MBV} \approx 28\ EC_{50}$). We did not performed experiments with concentrations in the range of the $C_{min}$ but the relation between $C_{max}$ and $C_{min}$ is available in the literature. For LMV, according to Kropeit *et al* [12], in healthy subject, for 120 mg of LMV $C_{min} = 0.1$ mg/L and $C_{max} = 2.5$ mg/L so $C_{min} = C_{max}/ 25$. As the pharmacokinetic of LMV is linear this relation can be used in our model. So, the minimal plasma concentration should be $C_{min(LMV)} = FC_{LMV}/25 = 1,000\ EC_{50(LMV)}/25 = 40\ EC_{50(LMV)}$. For MBV, according to Wang *et al* [9], in healthy subject, for 400 mg of MBV $C_{min(MBV)} = 3$ mg/L and $C_{max(MBV)} = 16.7$ mg/L so $C_{min(MBV)} = C_{max(MBV)}/ 5.6$. Therefore, the theoretical minimal plasma concentration of MBV, in the fetal compartment, in our model, should be $C_{min(MBV)} = FC_{MBV}/5.6 = 28\ EC_{50(MBV)}/5.6 = 5\ EC_{50(MBV)}$. In conclusion, theoretically, the

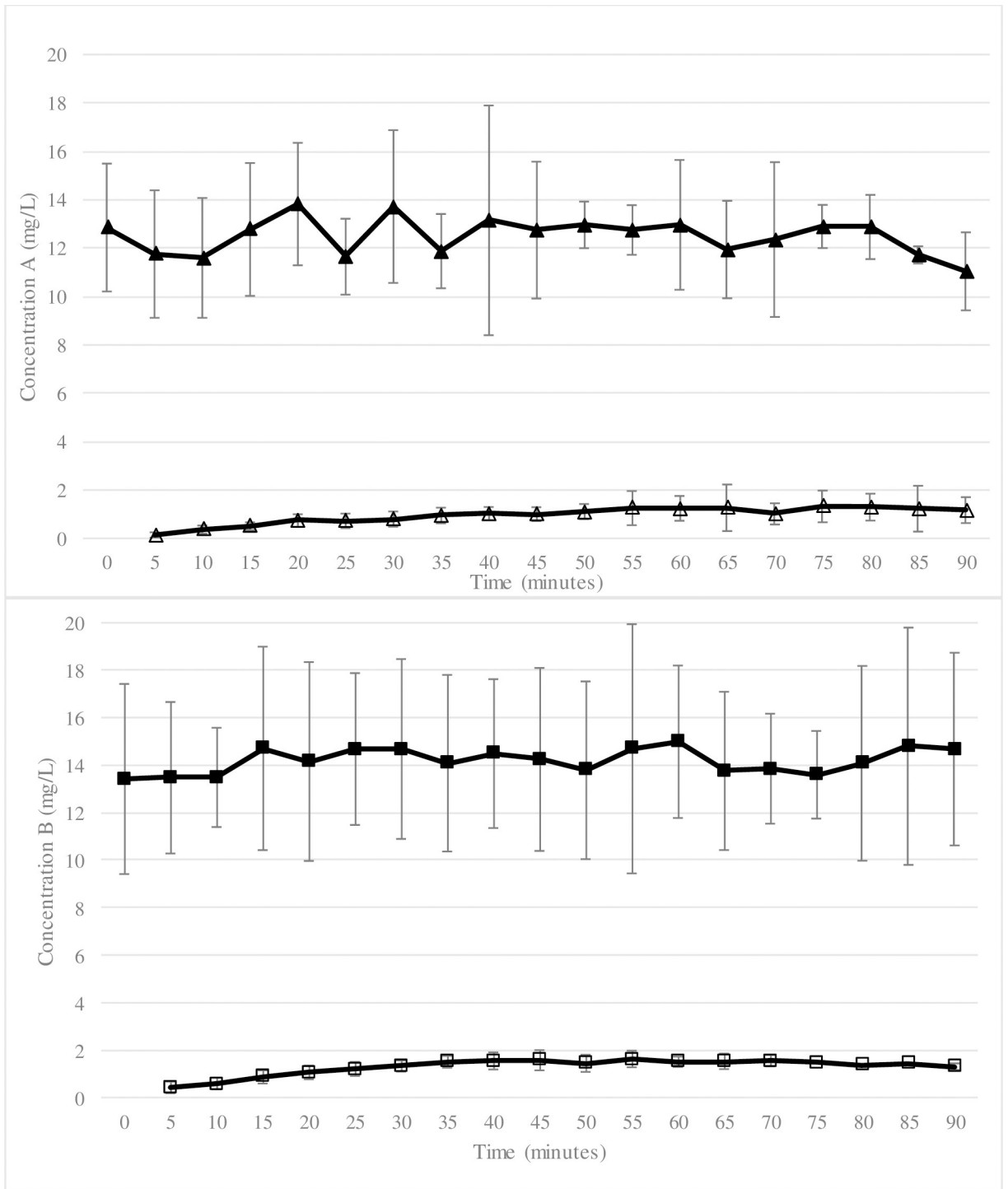

**Fig 1.** Mean Letermovir concentrations (Panel A) and mean Maribavir concentrations (Panel B) for all placentas. Panel A: Mean (± SD) maternal (full triangle) and fetal (empty triangle) concentrations of Letermovir (mg/L). Panel B: Mean (± SD) maternal (full square) and fetal (empty square) concentrations of Maribavir (mg/L).

concentration over a day should be, at any time, above the $EC_{50}$ for both molecules (between 40–1,000 $EC_{50(LMV)}$ and 5–28 $EC_{50(MBV)}$ for LMV and MBV respectively), so within the appropriate range to treat the fetus.

LMV is a moderate molecular weight (572.55 Da) and highly lipophilic (log P $_{oct/wat}$ = 4.58) molecule that acts via the viral terminase complex which cleaves DNA to package the genome into the capsid [13]. In vitro, LMV is a substrate of drug metabolizing enzymes: CYP3A, CYP2D6, UGT1A1, and UGT1A3, and transporters OATP1B1/3 and P-gp. 97% of the molecule is unchanged with no metabolites detected in the plasma [8]. The mean elimination half-life is 12 hours allowing an administration once daily (480 mg/d) [8]. In vitro, LMV is currently the most active molecule against CMV, with a very low EC$_{50}$ which surpasses the EC$_{50}$ of the Ganciclovir by more than 400-fold [14]. The safety profile of LMV is good, with minimal adverse effects and no documented hematologic toxicity or nephrotoxicity [8,15]. There are no human data on safety of LMV in pregnancy but no teratogenic effect nor maternal toxicity were reported in animal studies at up to 3-fold the recommended human dose, both in rats and in rabbits [8]. LMV has received approval from the FDA and European Medicines Agency, in November 2017, for the prevention of CMV infection and disease in adult allogeneic stem cell transplant patients [8].

MBV is a low molecular weight (376.23 Da) and lipophilic (log P $_{oct/wat}$ = 2.15) molecule that inhibits the UL97 viral kinase. It therefore opposes to viral DNA elongation, DNA packaging, and nuclear egress of encapsidated viral DNA [16]. MBV is rapidly eliminated, with a mean half-life in plasma of 3 to 5 h so it has to be administered orally twice-a-day [11]. The principal metabolite of MBV is 4469W94 (which is derived by N-dealkylation of MBV via CYP3A4) but this metabolite is pharmacologically inactive [9]. MBV also shows a favorable safety profile in both animals and human studies [9,17]. However, there is a lack of data about teratogenicity of MBV. MBV is being developed for treatment of CMV disease in patients with impaired cell mediated immunity. Two Phase III clinical trials have recently started in this population [18,19]. MBV has not yet been approved by the FDA.

Concentrations in fetal blood of both LMV and MBV are expected to be appropriate in terms of benefit/risk ratio. We have demonstrated that both anti-CMV drugs crossed the placenta at a low to moderate rate. These rates are concordant with the physico-chemical properties of the molecules with low to moderate molecular weight and a highly lipophilic profile. However, they are both highly bound to plasma proteins, mainly albumin (98% and 99% for MBV and LMV respectively) which is a significant obstacle to placental transfer since only the unbound fraction of the drug can cross the placenta membrane[20,21]. In our model we used 2 g/L of human albumin in the maternal and fetal solutions. This concentration is admitted as a standardized condition in this *ex vivo* model, by many experienced teams [5,22–24]. However, it is likely that fetal levels of free drugs *in vivo* will be lower than the values that were obtained in our perfusion experiments because most of the drug will remain bound to serum albumin in the maternal circulation. It is difficult and impractical to exactly mimic physiological protein binding during the perfusion experiment. We also think that using a human serum albumin is one of the strengths of our study since others did not add it at all to the perfusion solutions[25]. Optimal concentration is also difficult to define in this placental transfer model because significant changes in serum albumin occur during pregnancy. *In vivo*, when drugs are highly bound to plasma proteins, the amount of drug can significantly vary during pregnancy in both maternal and fetal compartments. Krauer B et al. showed that, maternal serum albumin concentrations ranged between 25 and 35 g/L and fetal serum albumin was much lower in early gestation, ranging from 7.5 to 16 g/L at 12–15 weeks. With advancing gestation, the authors noticed that there was a linear increase so that at 30 weeks both fetal and maternal serum albumin concentrations were in the same range and after 35 weeks the fetal concentrations exceeded the maternal by some 20% [26].

The issue of fetal toxicity cannot be addressed by our study. However, our results indicate that the fetus would be exposed to the drug, over a day, at concentrations between 40–1,000

$EC_{50(LMV)}$ for LMV and between 5-$28EC_{50(MBV)}$ for MBV. Whether these ranges could be toxic for a fetus is questionable. These data are theoretical, and it is difficult to predict what would exactly be the respective drug disposition in both maternal and fetal compartment *in vivo*.

The limitations of our model include that it does not account for fetal disposition nor for the accumulation of the drug in the amniotic fluid. Fetal drug metabolism has its own specificities, and these are disregarded in our *ex vivo* model. First, it has been shown that the zonation of xenobiotic metabolism is not localized in the same hepatic pattern in fetuses and adults [27]. Secondly, the enzyme activities are different especially in CYP activities where CYP3A7 is the predominant CYP enzyme in fetuses, while it is CYP3A4 in adults. Glucuronidation is also impacted with a limited activity of UDP-GT in the fetal liver leading to an immaturity of hepatic glucuronidation in fetuses and neonates. Full maturation may not occur for many years in the child [27,28]. The fetal pharmacology of LMV and MBV will be affected by these differences between fetal and adult hepatic elimination. Further investigations *in vivo* are required to evaluate the clinical implications.

Another limitation is the relatively small study sample of our study. However, studies published by other experimented teams also evaluated placental transfer of drugs on small studies samples [25,29–31]. We only used pure and titrated new drugs and human serum albumin, so the price of each experiment was very expensive. This high cost was the reason for not doing more experiments.

The issue of maternal toxicity is easier to appraise. LMV has a good safety profile in animal and in human studies [8,12,15]. Its oral administration once daily should facilitate acceptability by pregnant women. The US FDA pregnancy category has not been assigned yet. The French institute working on teratogenicity CRAT (Centre de Référence sur les Agents Tératogènes) [32] gave us a favorable opinion for the prospective use of LMV in pregnant women carrying a CMV infected fetus within a clinical trial. MBV is not available yet as it has not been approved by the FDA. Despite a satisfactory safety profile in the preliminary works, further studies are still needed.

The *ex vivo* cotyledon model allows to study the placental transfer of drugs that have not yet been tested in pregnant women, in standardized conditions and without ethical restrictions but it has several limitations. First, there is no guidelines as to respect the *ex vivo* infusion of placental human cotyledon, but merely one attempt by Mathiesen et al [33]. Second, it is a challenging procedure: half of our experiments were validated by an antipyrine FTR > 20%, which is a standard success rate [34]. Finally, it is technically available only for full-term placentas. As our final objective is to treat fetuses infected with CMV all along the pregnancy, the placental transfer of these drugs, earlier in pregnancy therefore remains somehow speculative.

The placentas used in our study were not infected with CMV. In the pathophysiology of cCMV infection, the placenta shows placentitis [2]. This may affect its architecture and therefore potentially also modify the transfer of molecules.

In conclusion, the new anti-CMV drugs LMV and MBV are promising in terms of efficacy, placental transfer and safety profile to be used to treat symptomatic infected fetuses with CMV. The latter aspect deserves caution, especially for MBV and phase-1 studies in infected pregnancies should be encouraged.

## Supporting information

**S1 Data.**
(XLSX)

## Acknowledgments

We thank the women who donated their placentas, the midwives who collected them and Dominique Duro (Assistance Publique-Hôpitaux de Paris, Hôpital Louis Mourier, Service de Gynécologie-Obstétrique, Colombes, France) for his tips on performing the *ex vivo* method of the human perfused cotyledon.

## Author Contributions

**Conceptualization:** Valentine Faure Bardon, Tiffany Guilleminot, Julien Stirnemann, Marianne Leruez-Ville, Yves Ville.

**Data curation:** Valentine Faure Bardon.

**Funding acquisition:** Yves Ville.

**Investigation:** Valentine Faure Bardon, Gilles Peytavin, Tiffany Guilleminot, Marianne Leruez-Ville, Yves Ville.

**Methodology:** Gilles Peytavin, Minh Patrick Lê, Marianne Leruez-Ville, Yves Ville.

**Supervision:** Marianne Leruez-Ville, Yves Ville.

**Validation:** Gilles Peytavin, Elisabeth Elefant, Marianne Leruez-Ville, Yves Ville.

**Writing – original draft:** Valentine Faure Bardon, Gilles Peytavin, Marianne Leruez-Ville, Yves Ville.

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
