## [Decision Letter · Decision Letter 0]

7 Feb 2020

PONE-D-20-01568

Placental transfer of Letermovir & Maribavir in the ex vivo human cotyledon perfusion model. New perspectives for in utero treatment of congenital cytomegalovirus infection.

PLOS ONE

Dear Dr Faure Bardon,

Thank you for submitting your manuscript to PLOS ONE. After careful consideration, we feel that it has merit but does not fully meet PLOS ONE’s publication criteria as it currently stands. Therefore, we invite you to submit a revised version of the manuscript that addresses the points raised during the review process.

We would appreciate receiving your revised manuscript by Mar 23 2020 11:59PM. To enhance the reproducibility of your results, we recommend that if applicable you deposit your laboratory protocols in protocols.io, where a protocol can be assigned its own identifier (DOI) such that it can be cited independently in the future. For instructions see: http://journals.plos.org/plosone/s/submission-guidelines#loc-laboratory-protocols

We look forward to receiving your revised manuscript.

Kind regards,

Sara Ornaghi, M.D., Ph.D.

Academic Editor

PLOS ONE

Journal Requirements:

"- Valentine Faure Bardon: No conflict

- Gilles Peytavin has received travel grants, consultancy fees, honoraria or study grants from various pharmaceutical companies, including Bristol-Myers Squibb, Gilead Sciences, Janssen, Merck and ViiV Healthcare.

- Minh Patrick Lê has received travel grants from Bristol-Myers Squibb and Janssen.

- Tiffany Guilleminot: No conflict

- Elisabeth Elefant: No conflict

- Julien Stirnemann : No conflict

- Marianne Leruez-Ville declares receiving financial support for meeting expenses from BioMerieux outside the submitted work.

- Yves Ville : reports non-financial support from Ferring SAS, non-financial support from Siemens health Care:, non-financial support from GE medical, outside the submitted work; "

Reviewers' comments:

Reviewer's Responses to Questions

**Comments to the Author**

1. Is the manuscript technically sound, and do the data support the conclusions?

Reviewer #1: Yes

Reviewer #2: Partly

2. Has the statistical analysis been performed appropriately and rigorously? 

Reviewer #1: N/A

Reviewer #2: Yes

3. Have the authors made all data underlying the findings in their manuscript fully available?

Reviewer #1: Yes

Reviewer #2: Yes

4. Is the manuscript presented in an intelligible fashion and written in standard English?

Reviewer #1: No

Reviewer #2: Yes

5. Review Comments to the Author

Reviewer #1: Faure V et al. examine placental transfer of two new CMV medications utilizing the human placenta cotyledon perfusion assay. This system is accepted and appropriate for pharma comparisons and evaluations of placental transfer. This study is of interest to the OB clinical and research community due to the continued danger that CMV poses to otherwise healthy pregnancies. The manuscript is well written (with exceptions cited below), experiments were done accordingly, and measurements appropriate using established modalities and the antipyrine control. However, there are important data elements missing, specifically in regards to clinical characteristics and technical details. The manuscript in its present state is suitable for publication with the following modifications:

1. Please include additional clinical characteristics: fetal sex, anesthesia, use of magnesium sulfate or other drugs/steroids, and if tested positive for Group B strep.

2. Correct inconsistencies in text and labeling. For example, in table one drug names are fully capitalized whereas the remaining tables only the first letter is capitalized. Please change so that only the first letter is capitalized when referencing these drugs. Finally table/figure titles should not be italicized.

3. Consolidate Figures 1 and 2 into a single figure and label the Letermovir graph as panel A and Maribavir as B. Also provide a figure legend that briefly describes the graphs that includes the labels (mean SD, triangle, and square) vs. stating them in the titles.

4. Ensure all words/labels are in English: for example in Table 2, pressure is spelled in French as pressian.

Reviewer #2: This manuscript consists of a study examining the maternal to fetal transfer of new CMV antiviral medications, Letermovir (LMV) and Maribavir (MBV), across the human placenta using a dual perfusion in vitro model. Authors demonstrated that both drugs crossed the placenta at a low to moderate rate. Their results indicate that the fetus would be exposed to the drug at concentrations between 40-1000 EC50 (LMV) and 5-28 EC50 (MBV). They signify clearly and correctly the limitations and strengths of the current study.

However there are several major limitations of the study:

1. Authors noted that both drugs are highly bound to plasma protein. In order to mimic maternal plasma protein concentrations authors should have added into the maternal circulation BSA in a concentration of 30gr/1 liter and not only 2gr/L. Since their main aims were to quantify the transfer of these drugs across the placenta, to my opinion this is an important obstacle of the study.

Although authors quote other study by Bapat P et. al 2015 who did not use at all albumin to their study, there are others who did it, specifically in a case of studying drug that known to be highly bound to plasma protein (Adverse placental effects of valproic acid: Studies in perfused human placentas. Rubinchik-Stern M et. al Epilepsia. 2018 May;59(5):993-1003)

2. This is a very small study sample, only 3 validated experiments for the MBV and 5 validated experiments for the LMV. Keeping in mind that very important conclusions are made based on the findings, it would be important to include a larger sample number for each drug

3. Measurements of placental viability and integrity during the periods of perfusion as HCG production, glucose consumption or oxygen transfer were not presented

6. PLOS authors have the option to publish the peer review history of their article (what does this mean?). If published, this will include your full peer review and any attached files.

Reviewer #1: No

Reviewer #2: No

---

## [Author Response · Author response to Decision Letter 0]

4 Mar 2020

Reviewer #1: Faure V et al. examine placental transfer of two new CMV medications utilizing the human placenta cotyledon perfusion assay. This system is accepted and appropriate for pharma comparisons and evaluations of placental transfer. This study is of interest to the OB clinical and research community due to the continued danger that CMV poses to otherwise healthy pregnancies. The manuscript is well written (with exceptions cited below), experiments were done accordingly, and measurements appropriate using established modalities and the antipyrine control. 

We thank the reviewer for this comment. 

However, there are important data elements missing, specifically in regards to clinical characteristics and technical details. The manuscript in its present state is suitable for publication with the following modifications:

1. Please include additional clinical characteristics: fetal sex, anesthesia, use of magnesium sulfate or other drugs/steroids, and if tested positive for Group B strep.

General clinical characteristics are now presented in table 1, in accordance with your request. 

2. Correct inconsistencies in text and labeling. For example, in table one drug names are fully capitalized whereas the remaining tables only the first letter is capitalized. Please change so that only the first letter is capitalized when referencing these drugs. Finally table/figure titles should not be italicized.

It was done. 

3. Consolidate Figures 1 and 2 into a single figure and label the Letermovir graph as panel A and Maribavir as B. Also provide a figure legend that briefly describes the graphs that includes the labels (mean SD, triangle, and square) vs. stating them in the titles.

It was done.

4. Ensure all words/labels are in English: for example in Table 2, pressure is spelled in French as pressian.

It was done.

Reviewer #2: This manuscript consists of a study examining the maternal to fetal transfer of new CMV antiviral medications, Letermovir (LMV) and Maribavir (MBV), across the human placenta using a dual perfusion in vitro model. Authors demonstrated that both drugs crossed the placenta at a low to moderate rate. Their results indicate that the fetus would be exposed to the drug at concentrations between 40-1000 EC50 (LMV) and 5-28 EC50 (MBV). They signify clearly and correctly the limitations and strengths of the current study.

We thank the reviewer for this comment. 

However there are several major limitations of the study:

1. Authors noted that both drugs are highly bound to plasma protein. In order to mimic maternal plasma protein concentrations authors should have added into the maternal circulation BSA in a concentration of 30gr/1 liter and not only 2gr/L. Since their main aims were to quantify the transfer of these drugs across the placenta, to my opinion this is an important obstacle of the study. Although authors quote other study by Bapat P et. al 2015 who did not use at all albumin to their study, there are others who did it, specifically in a case of studying drug that known to be highly bound to plasma protein (Adverse placental effects of valproic acid: Studies in perfused human placentas. Rubinchik-Stern M et. al Epilepsia. 2018 May;59(5):993-1003)

We agree with all comments concerning the concentration of albumin. 

In our model we used 2 g/L of human albumin in the maternal and fetal solutions like others experimented teams who work on the placenta perfusion. This concentration is admitted as a standardized condition in this ex vivo model, by many experienced teams. (1–4). 

Mandelbrot el al, when they analyzed the placental transfer of Rilpivirine, also a highly protein-bound (>99% ) molecule, used this concentration of 2g/L (2). Furthermore, he also used this concentration of albumin in a study published in Plos One last year about placental transfer of Dolutegravir “ 99% bound to plasma proteins, mainly albumin”(5). 

As previously mentioned, we think that using, even in a low concentration, human serum albumin is one of the strengths of our study since others did not add it at all to the perfusion solutions (6). Indeed, Bapat et al. mentioned that : “Perfusion experiments that include proteins in the perfusion medium are technically challenging because of the high viscosity of the perfusate. »

We agree that, it is likely that fetal levels of free drugs in vivo will be lower than the values that were obtained in our perfusion experiments, because most of the drug will remain bound to serum albumin in the maternal circulation. However, the human cotyledon perfusion model remains an ex vivo model with validated and standardized conditions but sometimes also far from the physiologically reality. Indeed, the procedure was carefully monitored and standardized to approach mimicking normal physiology and the FTR of antipyrine was monitored in order to control the integrity of the placental barrier. Differential protein binding between the fetal and maternal circulations is known to be a critical factor in the placental transfer of drugs. However, it is difficult and impractical to exactly mimic physiological protein binding during the perfusion experiment.. (7). 

However, even in low albumin conditions, the results obtained with this ex vivo placental perfusion model have been found to be consistent with in vivo findings for a number of compounds. (2,7). 

Finally, on a technical point of view, the increase of albumin concentrations in fetal solution might have some limitations:

- The dynamic agitation with magnetic rod or bubbling with gas in the respective vessels might frothing the maternal or fetal solutions, leading to risk of dismantling the peristaltic pumping

- the cost of human albumin is really expensive 

- the increase of albumin concentrations in maternal or fetal solutions to 5 or 10 g/L might be theoretically possible but the reservations expressed by the reviewer for 2 g/L are still arise regarding this point

2. This is a very small study sample, only 3 validated experiments for the MBV and 5 validated experiments for the LMV. Keeping in mind that very important conclusions are made based on the findings, it would be important to include a larger sample number for each drug

We agree with these comments however perfusion of a human cotyledon model is a challenging procedure. Only half of our experiments were validated by an FTR of antipyrine > 20 % which is a standard success rate(8). We only used pure and titrated new drugs and human serum albumin, so the price of each experiment was very expensive (approximatively 500 euros per placenta …) . This high-cost was one of the reason for not doing more experiments. 

Furthermore, studies published by other experimented teams also evaluated placental transfer of drugs on small studies samples : 

• raltegravir : maternal to fetal direction (n=3 placentas) (9)

• dolutegravir (n=5 placentas) (5)

• darunavir (n=5 placentas) (10)

• rivaroxaban: maternal to fetal direction (n=5 placentas) fetal to maternal direction ( n= 2 placentas) (6)

• oseltamivir (n=5 placentas) (11)

3. Measurements of placental viability and integrity during the periods of perfusion as HCG production, glucose consumption or oxygen transfer were not presented

Antipyrine transfer is a control of cotyledon’s viability. If Antipyrine FTR is < 20% the placenta is not validated. Most of laboratories use this antipyrine cut off of 20% to check on cotyledon’s viability (2,4,9,12–16) . 

On a technical point of view, we observed that the dynamic agitation with gas, in the respective vessels, might frothing the maternal or fetal solutions, leading to risk of dismantling the peristaltic pumping. This is why, like others experimented teams who work on the placenta perfusion, we have chosen not to use gas (1–3,5). In the same teams, HCG production and glucose consumption were also not recorded, Antipyrine transfer was sufficient to validate cotyledon’s viability. 

One of the limits of the placenta perfusion is that there are no standardized criteria between laboratories. Avoid toxicity associated with hyperoxia is an alternative procedure known and accepted (7). 

Furthermore, our parameters (fetal and maternal pressures and pH) were correctly maintained at physiological levels as mentioned in table 2. 

 References

 1. Ceccaldi P-F, Ferreira C, Gavard L, Gil S, Peytavin G, Mandelbrot L. Placental transfer of enfuvirtide in the ex vivo human placenta perfusion model. Am J Obstet Gynecol. avr 2008;198(4):433.e1-2. 

2. Mandelbrot L, Duro D, Belissa E, Peytavin G. Placental Transfer of Rilpivirine in an Ex Vivo Human Cotyledon Perfusion Model. Antimicrob Agents Chemother [Internet]. 5 janv 2015 [cité 21 nov 2015];59(5):2901‑3.

3. Gavard L, Gil S, Peytavin G, Ceccaldi P-F, Ferreira C, Farinotti R, et al. Placental transfer of lopinavir/ritonavir in the ex vivo human cotyledon perfusion model. Am J Obstet Gynecol. juill 2006;195(1):296‑301. 

4. Gavard L, Beghin D, Forestier F, Cayre Y, Peytavin G, Mandelbrot L, et al. Contribution and limit of the model of perfused cotyledon to the study of placental transfer of drugs. Example of a protease inhibitor of HIV: nelfinavir. Eur J Obstet Gynecol Reprod Biol. déc 2009;147(2):157‑60. 

5. Mandelbrot L, Ceccaldi P-F, Duro D, Lê M, Pencolé L, Peytavin G. Placental transfer and tissue accumulation of dolutegravir in the ex vivo human cotyledon perfusion model. PLOS ONE [Internet]. 13 août 2019 14(8):e0220323. 

6. Bapat P, Pinto LSR, Lubetsky A, Berger H, Koren G. Rivaroxaban transfer across the dually perfused isolated human placental cotyledon. Am J Obstet Gynecol. nov 2015;213(5):710.e1-6. 

7. Hutson JR, Garcia-Bournissen F, Davis A, Koren G. The Human Placental Perfusion Model: A Systematic Review and Development of a Model to Predict In Vivo Transfer of Therapeutic Drugs. Clin Pharmacol Ther [Internet]. juill 2011 [

8. Berveiller P, Gil S, Vialard F. Placental perfusion: interest and limits. J Matern Fetal Neonatal Med [Internet]. 3 juin 2017 ;30(11):1347‑8. 

9. Vinot C, Tréluyer J-M, Giraud C, Gavard L, Peytavin G, Mandelbrot L. Bidirectional Transfer of Raltegravir in an Ex Vivo Human Cotyledon Perfusion Model. Antimicrob Agents Chemother 1 mai 2016;60(5):3112‑4. 

10. Mandelbrot L, Duro D, Belissa E, Peytavin G. Placental transfer of darunavir in an ex vivo human cotyledon perfusion model. Antimicrob Agents Chemother. sept 2014;58(9):5617‑20. 

11. Berveiller P, Mir O, Vinot C, Bonati C, Duchene P, Giraud C, et al. Transplacental transfer of oseltamivir and its metabolite using the human perfused placental cotyledon model. Am J Obstet Gynecol 1 janv 2012 ;206(1):92.e1-92.e6. 

12. Vinot C, Gavard L, Tréluyer JM, Manceau S, Courbon E, Scherrmann JM, et al. Placental transfer of maraviroc in an ex vivo human cotyledon perfusion model and influence of ABC transporter expression. Antimicrob Agents Chemother. mars 2013;57(3):1415‑20. 

13. Huang H, Wang J, Li Q, Duan J, Yao Q, Zheng Q, et al. Transplacental transfer of oseltamivir phosphate and its metabolite oseltamivir carboxylate using the ex vivo human placenta perfusion model in Chinese Hans population. J Matern Fetal Neonatal Med [Internet]. 3 juin 2017;30(11):1288‑92.

14. Ceccaldi P-F, Mandelbrot L, Farinotti R, Forestier F, Gil S. Apports de la perfusion ex vivo du cotylédon humain dans l’étude du passage placentaire des médicaments. J Gynécologie Obstétrique Biol Reprod [Internet]. déc 2010 ;39(8):601‑5. 

15. Evain-Brion D, Berveiller P, Gil S. [Placental transfer of drugs]. Therapie. févr 2014;69(1):3‑11. 

16. Forestier F, de Renty P, Peytavin G, Dohin E, Farinotti R, Mandelbrot L. Maternal-fetal transfer of saquinavir studied in the ex vivo placental perfusion model. Am J Obstet Gynecol. juill 2001;185(1):178‑81.

---

## [Decision Letter · Decision Letter 1]

18 Mar 2020

PONE-D-20-01568R1

Placental transfer of Letermovir & Maribavir in the ex vivo human cotyledon perfusion model. New perspectives for in utero treatment of congenital cytomegalovirus infection.

PLOS ONE

Dear Dr Faure Bardon,

Thank you for submitting your manuscript to PLOS ONE. After a second round of review, we feel that your manuscript needs further revision (minor). Therefore, we invite you to submit a revised version of the manuscript that addresses the points raised during this second review process.

We would appreciate receiving your revised manuscript by March 27,2020. To enhance the reproducibility of your results, we recommend that if applicable you deposit your laboratory protocols in protocols.io, where a protocol can be assigned its own identifier (DOI) such that it can be cited independently in the future. For instructions see: http://journals.plos.org/plosone/s/submission-guidelines#loc-laboratory-protocols

We look forward to receiving your revised manuscript.

Kind regards,

Sara Ornaghi, M.D., Ph.D.

Academic Editor

PLOS ONE

Reviewers' comments:

Reviewer's Responses to Questions

**Comments to the Author**

1. If the authors have adequately addressed your comments raised in a previous round of review and you feel that this manuscript is now acceptable for publication, you may indicate that here to bypass the “Comments to the Author” section, enter your conflict of interest statement in the “Confidential to Editor” section, and submit your "Accept" recommendation.

Reviewer #1: (No Response)

Reviewer #2: All comments have been addressed

2. Is the manuscript technically sound, and do the data support the conclusions?

Reviewer #1: Yes

Reviewer #2: Yes

3. Has the statistical analysis been performed appropriately and rigorously? 

Reviewer #1: Yes

Reviewer #2: Yes

4. Have the authors made all data underlying the findings in their manuscript fully available?

Reviewer #1: Yes

Reviewer #2: Yes

5. Is the manuscript presented in an intelligible fashion and written in standard English?

Reviewer #1: Yes

Reviewer #2: Yes

6. Review Comments to the Author

Reviewer #1: Thank you for addressing the previous comments. The manuscript is improved and nearly suitable for publication. The only changes I request regard Figure 1. In my previous review I had asked that Figure 1 (Letermovir) and Figure 2 (Maribarvir) be merged into a single figure denoting panel A and B respectively, which the authors have done. However, the graphs were also merged (not requested), which creates confusion as a single graph represents only one panel but is labeled A and B. Please replace with the original two graphs and label Letermovir as A and the Marbarvir as B in vertical order. Also annotate the results section according, line 139 should read "Letermovir (Table 3, Fig 1A)" and line "Maribavir (Table 3, Fig 1B". Finally, please remove the abbreviations from the titles as indicated in my previous sentence. Once these minor changes are made, it is this reviewer’s opinion that the manuscript is ready for publication.

Reviewer #2: 1. Regarding author's response concerning the concentration of albumin. I accept their response. Yet a comment about this specific issue should be added, as they commented in here in their own wards: "It is likely that fetal levels of free drugs in vivo will be lower than the values that were obtained in our perfusion experiments, because most of the drug will remain bound to serum albumin in the maternal circulation. It is difficult and impractical to exactly mimic physiological protein binding during the perfusion experiment".

2. Regarding author's response about the small study sample. I accept their response. Please add a comment in the

study's limitations part, about the relative small study sample of the study.

3. Regarding the comment about measurements of placental viability and integrity during the periods of perfusion. I accept their response

7. PLOS authors have the option to publish the peer review history of their article (what does this mean?). If published, this will include your full peer review and any attached files.

Reviewer #1: No

Reviewer #2: No

---

## [Author Response · Author response to Decision Letter 1]

20 Mar 2020

Reviewer #1: Thank you for addressing the previous comments. The manuscript is improved and nearly suitable for publication. The only changes I request regard Figure 1. In my previous review I had asked that Figure 1 (Letermovir) and Figure 2 (Maribarvir) be merged into a single figure denoting panel A and B respectively, which the authors have done. However, the graphs were also merged (not requested), which creates confusion as a single graph represents only one panel but is labeled A and B. Please replace with the original two graphs and label Letermovir as A and the Marbarvir as B in vertical order.

It was done.

Also annotate the results section according, line 139 should read "Letermovir (Table 3, Fig 1A)" and line "Maribavir (Table 3, Fig 1B". 

It was done.

Finally, please remove the abbreviations from the titles as indicated in my previous sentence. 

It was done.

Once these minor changes are made, it is this reviewer’s opinion that the manuscript is ready for publication.

We thank the reviewer for this comment. 

Reviewer #2: 1. Regarding author's response concerning the concentration of albumin. I accept their response. Yet a comment about this specific issue should be added, as they commented in here in their own wards: "It is likely that fetal levels of free drugs in vivo will be lower than the values that were obtained in our perfusion experiments, because most of the drug will remain bound to serum albumin in the maternal circulation. It is difficult and impractical to exactly mimic physiological protein binding during the perfusion experiment".

It was done.

2. Regarding author's response about the small study sample. I accept their response. 

We thank the reviewer for this comment.

Please add a comment in the study's limitations part, about the relative small study sample of the study.

We added in the limitation part : “Another limitation is the relative small study sample of our study. However, studies published by other experimented teams also evaluated placental transfer of drugs on small studies samples(25,29–31). We only used pure and titrated new drugs and human serum albumin, so the price of each experiment was very expensive. This high-cost was the reason for not doing more experiments.”

3. Regarding the comment about measurements of placental viability and integrity during the periods of perfusion. I accept their response

We thank the reviewer for this comment.

---

## [Decision Letter · Decision Letter 2]

8 Apr 2020

Placental transfer of Letermovir & Maribavir in the ex vivo human cotyledon perfusion model. New perspectives for in utero treatment of congenital cytomegalovirus infection.

PONE-D-20-01568R2

Dear Dr. Faure Bardon,

We are pleased to inform you that your manuscript has been judged scientifically suitable for publication and will be formally accepted for publication once it complies with all outstanding technical requirements.

With kind regards,

Frank T. Spradley

Academic Editor

PLOS ONE

Reviewers' comments:

Reviewer's Responses to Questions

**Comments to the Author**

1. If the authors have adequately addressed your comments raised in a previous round of review and you feel that this manuscript is now acceptable for publication, you may indicate that here to bypass the “Comments to the Author” section, enter your conflict of interest statement in the “Confidential to Editor” section, and submit your "Accept" recommendation.

Reviewer #1: All comments have been addressed

Reviewer #2: All comments have been addressed

2. Is the manuscript technically sound, and do the data support the conclusions?

Reviewer #1: Yes

Reviewer #2: Yes

3. Has the statistical analysis been performed appropriately and rigorously? 

Reviewer #1: Yes

Reviewer #2: Yes

4. Have the authors made all data underlying the findings in their manuscript fully available?

Reviewer #1: Yes

Reviewer #2: Yes

5. Is the manuscript presented in an intelligible fashion and written in standard English?

Reviewer #1: Yes

Reviewer #2: Yes

6. Review Comments to the Author

Reviewer #1: Thank you for addressing the previous comments, it is this reviewers opinion that the manuscript is now suitable for publication.

Reviewer #2: (No Response)

7. PLOS authors have the option to publish the peer review history of their article (what does this mean?). If published, this will include your full peer review and any attached files.

Reviewer #1: No

Reviewer #2: No

---

## [Editor Report · Acceptance letter]

13 Apr 2020

PONE-D-20-01568R2 

Placental transfer of Letermovir & Maribavir in the *ex* vivo human cotyledon perfusion model. New perspectives for *in utero* treatment of congenital cytomegalovirus infection 

Dear Dr. Faure Bardon:

I am pleased to inform you that your manuscript has been deemed suitable for publication in PLOS ONE. Congratulations! Your manuscript is now with our production department. 

With kind regards,

on behalf of

Dr. Frank T. Spradley 

Academic Editor

PLOS ONE